# 'God helps those who help themselves'… religion and Assisted Reproductive Technology usage amongst urban Ghanaians

Rosemond Akpene Hiadzi, Isaac Mensah Boafo⬥*, Peace Mamle Tetteh

Department of Sociology, College of Humanities, University of Ghana, Accra, Ghana

* imboafo@gmail.com

**Data Availability Statement:** Data may be found at Mendeley repository using the following DOI:10. 17632/433nd5fhts.1 (https://data.mendeley.com/datasets/433nd5fhts/1).

## Abstract

Assisted Reproductive Technology (ART) is increasingly becoming a viable option for infertile couples in Ghana. There exists significant literature that explores the gender, legal, religious and socio-cultural implications of ART usage. In this paper, we expand the discourse on the nexus between religion and ART usage by looking at how the former is used as a frame of reference in the decision-making process, as well as how it is employed to explain treatment successes and failures. Irrespective of religious orientation, there was a general acceptance of ART by participants in the study-with exceptions only when it came to some aspects of the procedure. Even here, participants' desperate desire to have children, tended to engender some accommodation of procedures they were uncomfortable with because of their religious beliefs. Thus, in contrast to some studies that suggest religion as interfering with ART use, we posit that religion is not an inhibiting factor to ART usage. On the contrary, it is an enabling factor, engendering the agentic attitude of participants to find a solution to their infertility in ART; as well as providing the strength to endure the physical and emotional discomfort associated with the biomedical process of conception and childbirth. In this context, religion thus provides participants with a frame of reference to navigate the spaces between decision-making, treatment processes and outcomes, and attributions of responsibility for the outcomes whatever they may be.

## Introduction

The past four decades have seen a significant rise in the use of Assisted Reproductive Technologies (ARTs) in both developed and developing countries. In Ghana, the first successful IVF procedure was done in 1995 and since then, many privately owned fertility clinics have sprung up mainly in the capital, Accra and a few other urban areas. To date, there are about 19 fertility centres located mainly in Accra with a few in Kumasi, Takoradi and more recently, Ho. The increasing numbers of these treatment centres and their continuous existence mean that more Ghanaians are ready to use and are demanding ARTs in solving their infertility problems. In spite of the spread of these Western technologies in non-Western countries, only a few studies have been conducted in non-Western societies to examine the socio-cultural factors that

**Funding:** The authors received no specific funding for this work.

**Competing interests:** The authors have declared that no competing interests exist.

influence their use and their impacts in these societies. In Ghana, the few studies that have been conducted focused on the transnational operation of IVF clinics in Ghana [1, 2], as well as challenges experienced by clients undergoing ART treatment in Ghana [3]. In these studies, the authors describe the ways in which private fertility clinics in Ghana rely on medical professionals from Western countries to complement their local experts who albeit received training from Western countries to operate the clinics. Some of the challenges experienced by Ghanaian men and women accessing infertility treatment via ART include the high cost of treatment associated with it, the long distance to treatment centres, disturbances in their daily routine and work amongst others. Although religion permeates every aspect of the Ghanaian society, very little is known about the interplay between religion and ART treatment in Ghana. The current paper thus, seeks to address this lacuna in the extant opus by examining the role of religious beliefs in the acceptance or rejection of ARTs.

Indeed, religion and morality have been found to affect the acceptability of ARTs in many societies outside Ghana. Among the Christian population of Israel for instance, the biblical commandment to "be fruitful and multiply" has been the driving force behind the acceptability of ARTs both legally and socially [4]. This, coupled with the pronatalist culture of Israel stemming from the political desire to populate in order to occupy the vast lands seized from their Arab neighbours, has encouraged more births especially for the infertile through the use of ARTs [5]. ART services are thus highly subsidized to encourage more people to utilize them [6].

Issues of morality associated with the use of donor material for ART procedures also come to play among the Sunni Moslem population in Egypt. The use of donor semen is deemed inappropriate and equated with adultery, which is against existing religious and moral principles [7]. In other non-Moslem but traditional African societies, a similar belief was found to exist [8, 9]. Likewise, in Italy, the strong influence of the Catholic church has led to restrictions in certain aspects of assisted reproductive treatment for infertility such as cryopreservation and third-party donations [7]. However, Shia Moslem laws in Iran and Lebanon have permitted third party donations to facilitate conception using ARTs [10].

In this paper, an analysis is made of how respondents utilizing ART treatments understand and negotiate treatment with particular reference to any existing tensions and/or apparent equanimities that exist between religion and biomedicine. By so doing, we aim to contribute to the literature on the nexus between religion and modernity (with particular reference to ART treatment) as practiced, experienced, and understood by our Ghanaian respondents.

## The religious landscape in Ghana and the theory of agency

Prior to the advent of Christianity and Islam in Ghana via the work of colonialists/missionaries and trading activities respectively, Ghanaians practiced African Traditional Religion. In Ghanaian traditional religion, the hierarchy of authority which formed the basis of the religion consisted of different supernatural beings namely the Supreme Being, the nature gods, the ancestors and the other gods [11]. Each of them had a role to play in the fertility of the individual. Although the Supreme Being was regarded as the omnipotent, he was believed to work through the other deities since he was too powerful to be approached directly. The ancestors are believed to give children to the living for the continuation of the lineage. Believers therefore pray to the ancestors for fertility. Among the Akans, this is referred to as '*abawotum*' [12]. In addition, there are fertility gods that are believed to endow people with fertility. Anyone desirous of children therefore prays to such gods and offers the necessary sacrifices for obtaining such blessings from the gods. Children born through these means could then be named after these gods. Some trees are also believed to be inhabited by spirits which make people fertile

[13]. Cutting down such trees therefore meant invoking the wrath of the spirit dwelling in the tree and punishment for this act came in the form of infertility [13].

Furthermore, the ancestors and the gods are believed to bless 'good' people with many offspring. However, one form of punishment for evil was seen in the inability to conceive. As Caldwell and Caldwell [14] put it, traditional African religion affects fertility behaviour in the sense that, fertility is equated with virtue and spiritual approval whereas reproductive failure or cessation is regarded as a consequence of sin.

Apart from the role of these deities in the fertility of the individual, witchcraft is also believed to have the potential to negatively impact fertility. Witches are regarded as forces of darkness that cause evil and misfortune to others. One of such misfortunes could be infertility. Sackey [15], reports that the common belief in traditional Ghanaian society was that witchcraft was responsible for infertility. Okonofua, Harris et al. [16] in their research in Nigeria, also reported that witchcraft was often blamed for one's infertility. One way of combatting the effects of witches and to secure one's fertility was through the use of charms or amulets also referred to as *suman* in Akan [12]. These may be objects worn on or around different parts of the body and believed to be the repository of power derived from the gods or spirits.

In modern times, very few Ghanaians still practise African Traditional Religion (5.2%) with majority of them being Christians (71.2%) and a few (17.6%) being Moslems (GSS, 2012). With the advent of Christianity and Islam into the country, religious beliefs associated with infertility and its treatment have correspondingly shifted to align with Christian and Islamic beliefs. According to Christian beliefs, man was admonished by his creator to "be fruitful and multiply. . .." (Genesis 1:28). The primary value of a woman in biblical times lay in her chastity and after marriage, her reproductive ability. These may explain the desire to reproduce on one hand and the negative reactions that one suffers from members of the society if this God-given obligation is not fulfilled. Evidence from the Christian Bible suggests that God blesses his people with the capacity to reproduce when they act according to His will. "Your wife will be like a fruitful vine within your house. . ." (Psalm 128:3) and ". . .. none of your men or women will be childless. . .." (Deuteronomy 7: 14) are a few examples to that effect.

Furthermore, Islamic beliefs and practices associated with infertility and its treatment are similar to that of Traditional religious beliefs as well as Christian beliefs. In the Quran, it is written ". . .. we cause whom we will rest in the womb for an appointed term, then do we bring you out as babes. . .. . ." (Quran 22:5). 'We' here refers to Allah or the Moslem God. This also brings to the fore the issue of conception being controlled by a supernatural being namely Allah. Likewise, barrenness is seen as a decree from Allah, ". . .He leaves barren whom He wills. . . (Quran 42:50). However, this situation of infertility may not necessarily be a consequence of one's disobedience. Rather, Allah is seen as all-knowing and competent and His actions can therefore not be questioned. Such occurrences may therefore be seen as a test of faith.

According to the WIN/Gallup 2017 survey report which explored religious beliefs of over 66,000 people in 68 countries across the world, Ghana is ranked amongst the top ten most religious countries in the world with 94% of its populace professing to be religious. In addition, various scholars have pointed out the important role that religion plays in the life of the African. Opoku [17] notes that, religion is at the root of African culture and is the determining principle of African life; Africans are engaged in religion in whatever they do leading to Professor Idowu describing them as a people who are religious in all things [17]. Parrinder [18] states that Africans are incurably religious, whereas Mbiti [19] describes Africans as notoriously religious.

In addition to the knowledge of the percentage of Ghanaians that adhere to the three major religions prevalent in the country today and the realisation of the high levels of religiosity in

the country, is the emergence of the relatively new independent Christian churches or pente-costal-charismatic churches. These churches are becoming more appealing to the Ghanaian and by extension the African because according to Meyer [20], these churches seem to offer a more 'authentic,' Africanized version of Christianity than do the presumably Western oriented mainline churches. An important point worth noting here is that current pentecostal-charis-matic churches (PCCs) appear to derive their mass appeal at least, partly from propagating a 'complete break with the past'. The emergence of these new independent Christian churches brought about a shift from the erstwhile drive of converting traditional religious practitioners to Christianity to what is now generally a situation of converting orthodox Christians to the new form of Christianity which hinges on prosperity. According to this theology, financial prosperity, health and well-being are blessings from God to his faithful ones. The theology also challenges adherents to be in charge of their own destinies while at the same time demanding their rights from God as His children [21]. With its focus on the individual, the prosperity gos-pel in Ghana, especially, continues to erase traditional social arrangements that foster commu-nalism and collective endeavours. The effects of this orientation is the emergence of Assisted Reproductive Technologies in Africa which have come in to replace the more collective and communalistic strategies of solving infertility that existed in traditional African societies [22]. Another characteristic of these pentecostal-charismatic churches is the strong belief in the prowess of the head/lead pastor to intercede successfully on behalf of members to resolve vari-ous health, socioeconomic, and cultural problems.

Bearing in mind these differences in religious orientation of Ghanaians and the beliefs asso-ciated with them, this study sought to analyse the role of religion in the decision-making pro-cess towards the utilisation of ARTs in Ghana. For instance, do Ghanaian Christians and Moslems accessing ARTs differ in their attitudes towards ARTs? Do Ghanaian Catholics and Pentecostal-Charismatics accessing ARTs differ in their attitudes towards ARTs? These form the basis for this study.

The capacity of participants to act independently and make decisions of their own free choice to address their infertility reflects the sociological theory of agency. This theory is adopted as the sociological lens through which we appraise the attitude of participants in this study. The agency theory, first formulated in the academic economic literature, has been deployed in several disciplines including sociology to explain the motives, actions and interrelationships between actors in several social contexts. Gidden's model of agency and action emphasises the agency of human beings and the feature of 'intentionality' as a process. Such intentionality suggests that humans have definite goals during the course of their action. The 'reflexive monitoring' of action provides the rationalisation of action-i.e., humans capabilities to explain or give reasons for their conduct. A third concept of the agency theory of action, the 'motivation of action', speaks to the potential for action- i.e. individuals' capabilities to undertake the intended action. Thus, in any given situation, the grounds for action (rationalisation of action), motives for action (factors that engender action) and motivation for action (capabilities) are at play. These basic tenets of the agency action theory emphasise initiative, freedom, and creativity of individuals. In this paper, we adopt these concepts and ideas to demonstrate how couples, through intentionality, self-regulation and reflexiveness have sought to become parents through ART use irrespective of their religion. Thus, while participants' desire to become parents may have been influ-enced by the external pressures from their social systems, the agentic act to 'help oneself in order to be helped by God' permeated the decision-making process and the explanations participants gave to the outcomes of their use of ARTs.

## Methods

The study was conducted in a hospital setting in Accra, Ghana. The hospital was selected due to its approach to solving infertility, and thereby aimed to capture the behavior of Ghanaians seeking infertility treatment via Assisted Reproductive Technologies. The use of Assisted Reproductive Technologies (ART) is gradually increasing in Ghana, with more advanced options being based in private health facilities. The study site is one such privately-owned modern fertility clinic providing advanced ARTs, such as In-Vitro fertilization (IVF)—a procedure that involves the retrieval of eggs from a woman's ovaries and fertilizing them with sperm; and Intra Cytoplasmic Sperm Injection (ICSI)–a procedure that involves the direct injection of the sperm into the cytoplasm of the egg.

### Participants

Participants were restricted to 15 persons from married heterosexual couples. It also included 4 key informant interviews made up of an IVF specialist, an embryologist, a Catholic priest, and a pastor from a charismatic church. The IVF specialist and embryologist were purposively selected from the hospital that respondents were selected from. They were the key consultants in the hospital for anyone who needed fertility care. The Catholic priest was selected from a Catholic church close to the facility on the basis that clients may reach out to that church for their spiritual needs based on its proximity to the fertility centre. Finally, the pastor from the charismatic church was selected on the basis of the fact that, he organizes a yearly spiritual program for infertile couples seeking to have their own children under the theme: 'Operation 1000 babies'. The sampling was influenced by the need to understand the experiences of infertile married men and women from their own perspectives rather than achieving a representative sample.

### Procedure

Semi-structured in-depth interviews were the main mode of data collection. Respondents were mainly asked questions relating to the decision-making process towards seeking treatment and the role and reactions of their partners and family members. One-on-one interviews were conducted with participants in any one of these three languages, English, Twi and Ewe. All interviews transcribed directly into English.

During the interview process, 13 interviews were digitally recorded (with 2 participants declining). Notes were also taken during the interviews and read back to the respondents to confirm that they were consistent with the views of the respondents thus achieving respondent validation. Interviews lasted from 40 to 140 minutes (mean 90 minutes). Interviews were held in the facility, while the clients were either on admission or after an IVF or ICSI procedure. Two critical considerations for reflections of positionality in this study related to the researcher's outsider status deriving from gender and the status of not seeking help for infertility as were the participants. It was noticed that female participants were relatively much more relaxed (compared to their male partners) speaking to a fellow woman about their experiences of infertility and health seeking behavior, even if they perceived that the researchers did not have similar challenges of infertility. The men however appeared 'economical' with information which was attributed to the outsider factors named early on. To mitigate the implications of this for the study, the services of the embryologist who is a male was engaged, to sit in during the interviews and to ask some of the questions. This proved to be quite useful.

It was mainly women who took part in the study because interviews were conducted while these women were still on admission at the hospital after a successful embryo transfer. It was part of the requirement from the facility that clients stayed in the hospital five days after the

IVF or ICSI procedure to ensure they were resting as this they believe, could improve upon the success of the procedures. This time allowed them to be in a relaxed atmosphere and more receptive towards participating in the interviews since they had free time available. Although interviews were scheduled with their partners, they seldom made the time for it as they were constantly on the move attending to their normal duties and spent a short time on their visits to the hospital. They expressed the fact that, talking to their wives was enough since in their words 'my wife has said everything, and I do not have any new thing to say'. This is understandable given the Ghanaian cultural context whereby it is women who bear the brunt for infertility most and as such fits aptly into the maxim '(s) he who feels it, knows it'.

## Data analysis

Thematic analysis of data was employed following the guidelines provided by Braun and Clarke [23], namely, getting to know the data, generating codes, searching, reviewing and defining themes, and finally, writing up the findings [24]. After this, manual data coding began under appropriate potential themes, driven by theory and informed by the interview guide and research questions [23]. Where necessary, themes were modified to reflect the codes they contained and to reduce overlap. The analyses thus involved searching for repeated patterns of meanings.

## Ethical considerations

Ethical clearance was received from the Institutional Review Board of the Noguchi Memorial Institute for Medical Research, University of Ghana. Respondents who took part in the study signed and one participant who was semi-literate, thumb printed a consent form to show that they willingly took part in the study. To ensure anonymity, respondents names have been changed and identifying details removed.

## Findings

### Characteristics of participants

As presented in Table 1, the study involved 15 participants, all of whom were in heterosexual marriage unions. Overall, interviews were conducted with 13 females and 2 couples (interviewed together so recorded as 1 patient journey). Participants' age ranged between 20 and 59 with most of them between the 30–49 age bracket. Years of marriage to their current partner at the time of the study also ranged between 0 and 34 years with majority of them having been in their current union for less than 10 years. With respect to their infertility status, majority of the respondents [10] were experiencing primary infertility that is, had no child of their own and 5, secondary infertility, that is, were unable to have another child after one successful live birth. Respondents were mostly Christians [13] with 4 out of these being Catholics and the remaining 9 being Protestants. There were also 2 Muslims.

In terms of their socio-economic background, participants were mainly middle to high income earners with their occupation ranging from bankers, insurance managers, entrepreneurs, mining company workers and a few who were either living/working in the United Kingdom and other parts of Europe with their spouses. This is understandable as IVF treatment costs in Ghana are borne solely by the couple with no form of health insurance covering any aspect of the treatment.

**Religion and ART.** Religion was found to play three major roles in the quest for conception via Assisted Reproductive Technologies, namely, deciding whether or not to use ART; explaining treatment successes; and coming to terms with treatment failures.

**Table 1. Socio-demographic characteristics of respondents.**

| Item Description | Fertility clinic using ART (n = 15) |
|---|---|
| Males | 0 |
| Females | 13 |
| Couple | 2 |
| **Age of respondents**: | |
| 20–29 | 1 |
| 30–39 | 7 |
| 40–49 | 5 |
| 50–59 | 2 |
| **Years of marriage**: | |
| 0–4 | 3 |
| 5–9 | 5 |
| 10–14 | 4 |
| 15–19 | 1 |
| 20–24 | 1 |
| 25–29 | 0 |
| 30–34 | 1 |
| Primary Infertility | 10 |
| Secondary Infertility | 5 |
| **Religious Affiliation** | |
| Catholic | 4 |
| Charismatic/Pentecostal | 9 |
| Moslem | 2 |
| Source: Field interviews | |

## To use or not to use ART

The findings from the study suggested that among Christians, acceptability of ARTs may vary according to denomination. It was found, for instance, that some Catholic clients of fertility clinics had reservations towards IVF procedures based on the 'unnaturalness' of it and the fear of going against religious doctrines. Catholics accessing ART treatment expressed initial discomfort with the idea of artificial insemination and only went ahead with it after receiving assurance from the doctor that, embryos that are discarded are not viable. This puts them at ease because to them, discarding unused embryos is tantamount to abortion. Those who were unsure, found a way of seeking clarity from their priests prior to the commencement of the treatment. For instance, one of our respondents stated:

> *I went to my parish priest for special blessings just before coming in for the procedure. He prayed with me and gave me God's blessings. If he felt it was against Catholic beliefs, he wouldn't have gone ahead to bless me or am I lying? I don't think the church frowns on it and so neither do I. That is why I was able to ask God for his help so that everything will work out well.*

Respondents belonging to other Christian religious denominations, such as. the protestants did not express any reservations based on their religious beliefs towards the use of ARTs as treatment options for infertility. When asked if she felt being a pastor's wife meant she need not undergo infertility treatment using ARTs, 37-year-old Jennifer replied:

*No way!! It is even written in the bible that God helps those who help themselves. It is God who has given the doctor the wisdom to discover medicine to treat the sick. If today the doctor is telling me that he can use my eggs and my husband's sperms to give us a baby using scientific technology, why not? I don't see any problem here. In fact, I'm sure by now God is looking down at me and saying. . .Ah my daughter, you have done well! I will bless the work of your hands! (laughs with joy). I know he has done it for me already.*

At a Christian religious ceremony to pray for divine intervention for the fruit of the womb dubbed 'operation 1000 babies' organized by the International Prophetic Centre, a Charismatic church, participants were enjoined to bring along baby clothes of the required gender to church- blue for a baby boy and pink for a baby girl. According to the pastor,

*If you are here and you have been going to the hospital, looking for a baby and the doctor says you should come and do IVF, my sister, my brother, you better come along with two, three, four different clothes according to the color that you want. In fact, you can bring one blue, one pink, two blue one pink. . .. If you want twins, triplets, whatever number you want. Do not limit God, bring as many as you want. God can do it.*

Moslem clients were not left out of IVF and ICSI procedures. They were found accessing IVF and ICSI treatments in cases of both male and female factor infertility. One Moslem man encountered on one of the visits to a fertility clinic was in a polygynous marriage. Although he had seven children with his first wife, his second wife of ten years was childless. According to him,

*No, no, no, this has nothing to do with my religion. I want her to undergo an IVF procedure so that she can also have a child of her own. I love her very much and I want her to be happy. I know if she also gives birth, she will be happy. I want to bring honor to her. So we came to see the doctor so that he can help her. He said one of her fallopian tubes is blocked but through IVF, she can get pregnant. I don't believe there is anywhere in the Quran that Allah speaks against this. No!*

After the decision has been taken to access ART treatment, male clients are required to produce semen samples by either masturbating or having sexual relations with their spouses within the hospital. Key informant interviews with the embryologist revealed that Moslem clients expressed more discomfort with the idea of masturbation which led to some withdrawing from treatment without considering the possibility of the production of semen via sexual intercourse with their partners within the hospital. According to him,

*Some men do not like the idea that they have to masturbate to produce their semen for analysis. I've realized it is mostly so with our Moslem clients. I am not a Moslem but I think there must be something about masturbation written in the Quran because as soon as I tell them, they shake their heads and tell me no, they cannot do it.*

In addition, the data revealed that, Moslems accessing IVF treatment did not subscribe to the use of donor material. This is because it was against religious doctrines to involve a third party in the fertilization process. This was seen as adultery. Key informants revealed that it is in very rare cases, Moslem clients accessed donor IVF procedures.

## Explaining treatment successes—God is the ultimate healer

Treatment successes were understood not only in terms of the capabilities of science and technology but more importantly in religious terms as the ultimate hand of God being at work.

The role of a supreme being in the success of IVF/ICSI treatments is seen in the local names that respondents indicated they would give to such children. Two respondents revealed they would name their children Nukunu—an Ewe name meaning 'a wonderful thing/creation'. This aptly expresses the sentiments of the expectant parents regarding the means of conception. Others had also selected Ghanaian names with religious undertones to express their sentiments towards God's role in the process of achieving desired conception. Ghanaian names such as Seyram (God has blessed me), Aseye (Praise Him), Nyamekye (God's gift) and Nhyira (God's blessings) were some of the names that respondents had chosen for their children.

A divine hand is also seen to be at play throughout every step of the process beginning from the medications taken in preparation for the procedure, through conception to successful gestation and delivery. Aside having undergone a successful IVF procedure and having carried the baby in her womb for 29 weeks, Akuvi's experience at labour and successful vaginal delivery was the icing on the cake for her desire to experience motherhood. At 57, the doctor had advised her to have a caesarean birth but she had convinced him she would try and give birth naturally. She narrated her experience with such joy and in such detail. During her week-long stay at the hospital following the birth, she was easily noticeable as she was often found walking up and down the corridors between the neo-natal intensive care unit where her pre-mature baby was receiving care. She would be seen waving her white handkerchief amidst songs of praises to God for his goodness. According to her,

> "...it is God ooo, it is God. At long last, my dream has come to pass, my dream has come to pass, it is all in the past now".

Likewise, multiple pregnancies and births were interpreted as additional blessings from God. Despite the fact that transfers of more than one embryo at a time is the norm (except in few cases where only one embryo is available for transfer), the ability for more than one embryo to be implanted is regarded as divinely coordinated. For Adzo, after ten years of a childless marriage, God had finally blessed her not with one, but two babies. By so doing, He has brought happiness to her and shamed all those who were ridiculing her because of her childlessness. The sentiments of Adzo is thus a representation of how participants attributed treatment successes to God and how multiple births were considered to be extra blessings from God.

## Coming to terms with treatment failures

When procedures are not successful, clients again fall on religion as an explanatory model. However, the supernatural entity (God) is never seen as opposed to their desire to have children. Rather, it is blamed on circumstances such as timing. Although they could not explain why they were unsuccessful at that particular point in time, as one of the respondents put it, "God's time is the best, His ways are not our ways". This meant one had to keep trying until the right time when God chooses to 'endorse' the conception and birth. Laureen, who miscarried each one of her triplets after almost eight weeks of pregnancy expressed a similar sentiment. Although it was a very painful experience, since she lost them, one after the other, she expresses joy at the opportunity to also feel what it is like to be pregnant even if for a short time. According to her,

> "Only God knows why I lost all three of them and in such a manner. Especially Lisa/ Seyram, the last one. I was so sure that she would survive because each time the doctor did a scan, we

*could hear her heartbeat. I kept saying, God, as for this one, do not take her away. But in the long run, the doctor discovered she was growing outside my uterus and I could have died if the baby had survived. That is why I was bleeding so much. So I believe that, despite the fact that they all didn't survive, at the end of the day, God knows best that is why I am still alive today".*

Service providers were never blamed for treatment failures. This may probably be because, the doctor had told the clients over and over again that,

We (the specialist together with his team) have done everything that is humanly and scientifically possible and can only do so much. The rest is up to God.

Clients and service providers thus interpreted and understood treatment failures in scientific and more importantly in religious terms as well. As indicated by the pastor:

*Is it everyone who will give birth? There are examples of people in the Bible who never had children. Everyone has a purpose on this earth and for all you know, God's plan for such people is different. It could be that, they have been brought here to take care of the many orphans out there who do not have anyone to take care of them.*

## Discussion

The main objective of this study was to explore the role of religious beliefs on the utilization of ART treatment as a solution to infertility. The religious beliefs of clients showed a general acceptance of the procedure. It is often regarded as the work of God through man. The IVF specialist was seen as one who was utilizing the knowledge that God had endowed man with to cater for their needs. The Catholic Church is known to be against anything artificial that aims at interfering with the process of reproduction. Catholics do not subscribe to the use of contraceptives to prevent conception and are also against abortion equating it to the sin of murder (www.catholic.com/tracts/abortion-birth-control). In addition, the manipulation of human embryos ex-vivo and the discarding of excess embryos are against Catholic doctrines [25]. The Catholic Church's position on this explains why some (albeit very few) felt uncomfortable with some aspects of the process. Their change in acceptability of the procedure was borne more out of the need to have children than religious doctrines.

In her study of the intersections between religion and modernity, Roberts [26], discovered that, in the strong Catholic state of Ecuador, there was a stable separation between the church and the state in the area of IVF treatments. As such, despite the existence of strong Catholic sentiments for both patients and practitioners, these did not influence the acceptance of IVF in a negative way. In addition, the services of religious leaders were employed to aid in fertilisation and implantation. Despite the fact that Ghana does not possess the identity of being a Catholic state, such similarities in the separation of church doctrines and actual practice in the field of IVF treatments is useful in providing an understanding of the synergies between religion and science in varying contexts.

Respondents belonging to other Christian religious denominations did not express any reservations towards the procedure based on their religious beliefs. This may be because, other Christian denominations are not as dogmatic in orientation in the area of reproduction and the non-interference of human agents in the process. Perhaps, for all groups of believers, their high levels of education could also explain their openness to new scientific information.

Furthermore, religious explanations that clients accessing IVF and ICSI give are often evident at all three stages of treatment namely before treatment, after a successful procedure and

also after an unsuccessful procedure. Before treatment, clients often explained their decision to undergo the procedure with the common Christian phrase that says "God helps those who help themselves". This means that they did not see the scientific procedure as in opposition to God's will. Religion is therefore not a conflicting system but a complementing one. Religious faith is also important in providing a source of strength to cope with the emotional and physical pains associated with treatment.

In Moslem marriages whereby polygyny is practiced, female factor infertility can be resolved simply by marrying another wife. However, this is not always the case. This means that religious rules permitting polygyny do not always prevent Moslems from accessing IVF treatment. In other words, female factor infertility as evidenced in the case of the Moslem man in a polygynous marriage who was seeking treatment for his second wife of 10 years, posed no threat to the marriage for the woman.

For the Moslem also, masturbation is regarded as sin since it is tantamount to having an extra marital affair.

In addition, the data revealed that, Moslems accessing IVF treatment did not subscribe to the use of donor material. This is because it was against religious doctrines to involve a third party in the fertilization process. This was seen as adultery and is in tandem with findings amongst the Sunni Moslems where it was considered illegal to use donor material [10]. However, the extent to which such religious doctrines are strictly adhered to is also of interest. Key informants revealed that, in very rare cases, Moslem clients accessed donor IVF procedures. This attitude could again be attributed to the greater need to have a child.

Kahn [27] in her study of Orthodox Jews in New York who were undergoing infertility treatment found that a common Jewish principle influenced the way both doctors and patients interpret treatment successes and failures. This Jewish religious principle enjoined believers to exert the most effort at whatever task they were performing in order to achieve a desired outcome. However, God determined the ultimate success of that effort.

According to Adjah [28], the name that Ghanaians give to their children is determined by a number of factors including the circumstances surrounding the birth of the child as well as the parents' beliefs and ideas. Names based on parents' beliefs and ideas are otherwise referred to as allusive names. Allusive names do not have a direct bearing on the child but rather portray the parents' ideas about man's relationship with God amongst other things [29]. The origin of such names can be likened to the coming of Christianity [30]. The Ghanaian thus ended up responding to this new religion by adopting Christian names in local languages. These two factors surrounding the naming of a child in Ghana explains the behaviour of the participants in this study. It is apparent that, their choices of names for their children that will be borne out of a successful IVF/ICSI procedure stems from the circumstances surrounding the birth as well as their beliefs and ideas about God.

## Limitations

As with most qualitative studies, this study does not seek to generalise that, amongst Ghanaians, religion is supportive of ART usage. The study focused on users of ART and how their decisions to use ART took into account their religious beliefs. The study is thus limited in scope as it excludes infertile persons who are not seeking treatment. Further studies may thus be needed to document if our findings hold true for infertile persons who are not seeking ART treatment. Again, this study did not explore how individuals not seeking ART might compare with ART users at the study site i.e. whether the religious background of infertile couples impact perceived uptake of fertility services at fertility clinics/hospitals thus the need for further studies in that regard.

## Conclusion

In analyzing the effects of Western biomedical treatment options for the infertile and their acceptability and utilization in the Ghanaian context, varying considerations emerged. These were influenced by existing social, cultural and religious beliefs. Socio-culturally, based on the overarching desire to be biological parents, clients navigated their way around treatment options in such a way as to reduce dissonance to the barest minimum. The different worlds of science and religion were also found to be necessary synergies that supplement/complement one another rather than being in opposition.

In sum, the major beliefs associated with infertility and its treatment in Ghana point to the belief in fertility being orchestrated by a supernatural entity. For that matter, believers rely on these deities and their representatives in ensuring their fertility. In times when the potential for fertility is challenged, they again draw on religious beliefs and practices in the hope of obtaining a cure for their infertility.

## Acknowledgments

The authors are grateful to the staff of the fertility hospital and the various respondents as well as the University of Ghana NGAA Carnegie Project for their support in carrying out this study.

## Author Contributions

**Conceptualization:** Isaac Mensah Boafo, Peace Mamle Tetteh.

**Data curation:** Rosemond Akpene Hiadzi, Peace Mamle Tetteh.

**Formal analysis:** Rosemond Akpene Hiadzi, Peace Mamle Tetteh.

**Methodology:** Rosemond Akpene Hiadzi, Isaac Mensah Boafo.

**Writing – original draft:** Rosemond Akpene Hiadzi.

**Writing – review & editing:** Isaac Mensah Boafo, Peace Mamle Tetteh.

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
