## [Decision Letter · Decision Letter 0]

13 Apr 2021

PONE-D-20-36012

‘God helps those who help themselves’… religion and ART usage amongst urban Ghanaians

PLOS ONE

Dear Dr. Boafo,

Thank you for submitting your manuscript to PLOS ONE. After careful consideration, we feel that it has merit but does not fully meet PLOS ONE’s publication criteria as it currently stands. Therefore, we invite you to submit a revised version of the manuscript that addresses the points raised during the review process.

Both reviewers have pointed out strengths in this manuscript. Yet, they both have main critical concerns that need to be carefully addressed before reconsidering this paper for publication. 

We look forward to receiving your revised manuscript.

Kind regards,

Sara Rubinelli

Academic Editor

PLOS ONE

Journal Requirements:

We note that you have indicated that data from this study are available upon request. PLOS only allows data to be available upon request if there are legal or ethical restrictions on sharing data publicly. For more information on unacceptable data access restrictions, please see http://journals.plos.org/plosone/s/data-availability#loc-unacceptable-data-access-restrictions.

2a) If there are ethical or legal restrictions on sharing a de-identified data set, please explain them in detail (e.g., data contain potentially sensitive information, data are owned by a third-party organization, etc.) and who has imposed them (e.g., an ethics committee). Please also provide contact information for a data access committee, ethics committee, or other institutional body to which data requests may be sent.

2b) If there are no restrictions, please upload the minimal anonymized data set necessary to replicate your study findings as either Supporting Information files or to a stable, public repository and provide us with the relevant URLs, DOIs, or accession numbers. For a list of acceptable repositories, please see http://journals.plos.org/plosone/s/data-availability#loc-recommended-repositories.

We note you have included a table to which you do not refer in the text of your manuscript. Please ensure that you refer to Table 1 in your text; if accepted, production will need this reference to link the reader to the Table.

Reviewers' comments:

Reviewer's Responses to Questions

**Comments to the Author**

1. Is the manuscript technically sound, and do the data support the conclusions?

Reviewer #1: Partly

Reviewer #2: Yes

2. Has the statistical analysis been performed appropriately and rigorously? 

Reviewer #1: N/A

Reviewer #2: N/A

3. Have the authors made all data underlying the findings in their manuscript fully available?

Reviewer #1: No

Reviewer #2: No

4. Is the manuscript presented in an intelligible fashion and written in standard English?

Reviewer #1: Yes

Reviewer #2: Yes

5. Review Comments to the Author

Reviewer #1: Review

This manuscript describes the attitudes of couples in Ghana using Assisted Reproductive Technology to become pregnant, and how their religious affiliation affects these attitudes. This is a highly important but sensitive topic, which makes it hard to systematically collect data – I think the author do the subject justice with their data analysis and their differentiated discussion. I enjoyed reading the ethnographic descriptions, and I think they capture the struggles of these couples while undergoing ART. I therefore think this work deserves publication. To improve it even further, I would recommend bolstering the manuscript with additional information in places and add more differentiated reflections on different factors that might have influenced the attitudes of the participants – mainly on participant social class and the ethnographer’s own role. It would be good to discuss how representative the sample is for Ghana. I will describe recommended changes below.

L.21 – I would not start the Abstract with ‘ART’ because many readers will not know what this is right away.

L.35 – The manuscript jumps between ‘decision-making’ and ‘decision making’.

L.49 – It would be good to extend on these studies about ART in Ghana and the challenges experienced by couples – this is important in setting the stage for the whole article, so adding one or two sentences would be well justified.

L.56 – The article treats ‘non-Western’ as a unified block, but this is clearly not the case when it comes to morality and reproduction. There is also subsequently only a discussion of Abrahamic monotheistic religions. What about polytheistic or atheistic countries (especially India and China should have extensive use of ARTs)? What about different communities within other countries (e.g., contrasts between Catholic and Protestant believers in South American countries)? For example, later in the text there is a sentence about intersectionality in Ecuador. It feels like the argument the authors are trying to make in the Discussion could be improved if the manuscript is embedded better in a global context – similar processes will be at work around the world, but currently we only learn about a handful of countries.

L.161 – This whole paragraph (‘On the whole,…’) seems to be more interpretation based on the data than actually part of the Introduction, so it feels out of place. There also seems to be some value judgment in this – when the authors describe the new independent churches as ‘not being an opium’, this seems like a criticism of traditional churches. I would scrap this paragraph and replace it with a description of the research hypotheses based on the preceding discussion of different religious attitudes – would we expect differences between different religious groups? It would also be good to discuss upfront the methodology of this study – it is hard to get quantitative data on this topic so ethnography is a very good way forward, but it comes with a number of problems in this case (selection bias, the role of the ethnographer; see below) and it would be good to describe why this method was considered appropriate nevertheless.

L.169 – It would be useful to get more information in the text about the ‘sociological theory of agency’, what it entails, and how it is employed here. It feels like this piece of information is dropped on the reader, without citation or clear explanation, even though it seemingly is the main explanatory theory used in the manuscript. Has this theory been employed for ART decisions before, or the actions of religious people in moral contexts? While the Introduction is rich with descriptions of the religious landscape in Ghana, the formal framework under which the authors interpret is mainly implied.

L.180: Again, it clearly is very hard to get these data, and the authors have done a fantastic job getting those interviews. However, the fact that all the interviews were collected in one facility and only with couples who have already decided to use ART causes a host of problems. The main two are:

a) Couples in Accra who can afford ART might have different religions, but they are more urbane and probably more wealthy than the average couple in Ghana, and they probably have a higher level of education and access to fertility-related information (I make the assumption here that IVF is expensive in Ghana). So, the sample is varied when it comes to religion, but quite homogenous with regard to other factors. The assumption underlying the manuscript is that religion matters when it comes to these decisions and that couples interpret their religion as a way to improve their fertility, but what if that is not the case for less wealthy or more rural populations? In that case, the manuscript does only reflect a very particular slice of Ghana’s society, and that needs to be discussed and the discussion needs to be more nuanced.

b) By only choosing couples who decided to undergo ART, the authors face a selection bias: only couples whose religious feelings and communities allowed them to do ART are even interviewed, and they will obviously have a positive attitude to the subject. So, to make the conclusion that religion is a supportive factor in ART might be premature: we do not currently know what keeps other couples from attempting IVF. It might be that 95% of people reject ART based on their religion, but because the authors only interview 5%, they will not notice this in their interviews. Again, there is not much the authors can do about this, given the difficulties of getting data, but this possibility needs to be discussed and where possible, additional evidence needs to be presented to rule this out.

L.185 - I would include a brief description of what IVF and ICSI stand for, because they have different religious implication later on.

L.186 - How were key informants selected?

L.192 - The authors need to discuss their own role in this process more and potential biases that could have been introduced into the results by their own positionality. For example, later we learn that almost only women sat down for interviews, and men only as part of couples. This is worrisome and probably to do with cultural aspects and the role of the interviewer, but again this needs a nuanced discussion.

L.215 - Does the thumbprint indicate that some participants were not literate? It feels like that is a matter worth discussing, as this could bias their responses (e.g., more dependent on community leaders to determine their attitude, more dependent on clinic staff to detail the moral implications for them).

L.227 - Could you explain what primary and secondary infertility are in this context?

Table 1: There are many striking characteristics in this table that need to be explained in a more differentiated way, I think. First, the large age and relationship history ranges: presumably, these couples differ dramatically in why they attempt ART, how often they have tried in the past, societal pressures, and so on. The article is not specifically designed to address these questions, but presumably a couple that has been trying for pregnancy for 30 years might have a very different outlook on what is permissible than one that has been trying for 2 years, and might get very different advice from their community.

Analyses and Discussions: Based on the mentioned uncertainties introduced by the method, I think the manuscript would benefit from a more detailed discussion of how representative the sentiments of the couples and their religious attitudes are for Ghana, and whether the sample selection process might have biased the interviews towards those who interpret their religion in pro-fertility way. Also, especially for the Pentecostal religions, where most couples will be converts, it is possible that participants actually joined the particular church that was most positive towards their fight with infertility and promised them to solve it.

Reviewer #2: An important question is whether the study meets the PLoS ONE data policy; the authors note limitations on the availability of the data. If those restrictions do not meet PLoS ONE data requirements, then publication criteria would preclude publishing the manuscript. Apart from that, the manuscript offers a well-structured, substantive, qualitative evaluation of a sample of Ghanians seeking and being treated with ART and how that relates to their religious beliefs. The introduction and discussion sections situate the study well, and reference both relevant findings and concepts in Ghana as well as other comparative settings. The take-away empirical points are made clearly, with illustrative quotations also useful.

The manuscript could benefit from a modest copyediting process to clean up some occasional stray commas and minor language elements, but overall the manuscript is quite readable and easy to follow. As far as other comments related to modest potential edits, please comment how common ART is in Ghana, what form(s) of ART were employed by study participants (a table in the results just points to primary and secondary fertility without providing more information), and how those services are paid for (the manuscript says in a private facility, so I am guessing this is out-of-pocket which in turn warrants brief comment on the socioeconomic backgrounds of those who can access such services).

Are there any indications how representative this particular sample's findings are relative to other Ghanian ART service providers? Since all but 2 patients who were participated were women, please comment on the general ART services and demographics of those who use those services at the facility (e.g., are most patients women, or might the present sample say something more about difficulties of recruiting males who use ART?).

The sample of patients only includes, by nature of the study design, those who have used ART. The study does not include patients who, perhaps by virtue of religious beliefs, elected not to receive ART. Based on the interviews with the four key informants, did they offer any insights about how commonly patients decline ART due to religious beliefs? If so, please share more from that vantage in the results.

6. PLOS authors have the option to publish the peer review history of their article (what does this mean?). If published, this will include your full peer review and any attached files.

Reviewer #1: No

Reviewer #2: No

---

## [Author Response · Author response to Decision Letter 0]

10 Jul 2021

We would like to thank the reviewers for taking time off their schedules to provide us with comments that has helped in strengthening the paper. We have addressed all comments adequately in the revised manuscript. We provide summaries of how we have responded to the comments below and referred to the appropriate section where such responses could be seen. Please note that these references are made in relation to the marked copy and not the clean copy. Thank you.

Reviewer #1: Review

This manuscript describes the attitudes of couples in Ghana using Assisted Reproductive Technology to become pregnant, and how their religious affiliation affects these attitudes. This is a highly important but sensitive topic, which makes it hard to systematically collect data – I think the author do the subject justice with their data analysis and their differentiated discussion. I enjoyed reading the ethnographic descriptions, and I think they capture the struggles of these couples while undergoing ART. I therefore think this work deserves publication. To improve it even further, I would recommend bolstering the manuscript with additional information in places and add more differentiated reflections on different factors that might have influenced the attitudes of the participants – mainly on participant social class and the ethnographer’s own role. It would be good to discuss how representative the sample is for Ghana. I will describe recommended changes below.

L.21 – I would not start the Abstract with ‘ART’ because many readers will not know what this is right away.

Abstract has been revised to read Assisted Reproductive Technology 

L.35 – The manuscript jumps between ‘decision-making’ and ‘decision making’.

The manuscript is now consistent with the use of decision-making

L.49 – It would be good to extend on these studies about ART in Ghana and the challenges experienced by couples – this is important in setting the stage for the whole article, so adding one or two sentences would be well justified.

This has been done. See lines 52-57

L.56 – The article treats ‘non-Western’ as a unified block, but this is clearly not the case when it comes to morality and reproduction. There is also subsequently only a discussion of Abrahamic monotheistic religions. What about polytheistic or atheistic countries (especially India and China should have extensive use of ARTs)? What about different communities within other countries (e.g., contrasts between Catholic and Protestant believers in South American countries)? For example, later in the text there is a sentence about intersectionality in Ecuador. It feels like the argument the authors are trying to make in the Discussion could be improved if the manuscript is embedded better in a global context – similar processes will be at work around the world, but currently we only learn about a handful of countries.

This has been rectified. The sentence now reads, “in many societies outside Ghana…”. See lines 63-64 

L.161 – This whole paragraph (‘On the whol,…’) seems to be more interpretation based on the data than actually part of the Introduction, so it feels out of place. There also seems to be some value judgment in this – when the authors describe the new independent churches as ‘not being an opium’, this seems like a criticism of traditional churches. I would scrap this paragraph and replace it with a description of the research hypotheses based on the preceding discussion of different religious attitudes – would we expect differences between different religious groups? It would also be good to discuss upfront the methodology of this study – it is hard to get quantitative data on this topic so ethnography is a very good way forward, but it comes with a number of problems in this case (selection bias, the role of the ethnographer; see below) and it would be good to describe why this method was considered appropriate nevertheless.

The paragraph has been deleted and a description of the main research question have been included as suggested. See lines 176 to 181

L.169 – It would be useful to get more information in the text about the ‘sociological theory of agency’, what it entails, and how it is employed here. It feels like this piece of information is dropped on the reader, without citation or clear explanation, even though it seemingly is the main explanatory theory used in the manuscript. Has this theory been employed for ART decisions before, or the actions of religious people in moral contexts? While the Introduction is rich with descriptions of the religious landscape in Ghana, the formal framework under which the authors interpret is mainly implied.

The issues raised here have been addressed. The key tenets of the agency theory employed in this paper have been stated and explained, as has the relevant citation. See lines 187-201

L.180: Again, it clearly is very hard to get these data, and the authors have done a fantastic job getting those interviews. However, the fact that all the interviews were collected in one facility and only with couples who have already decided to use ART causes a host of problems. The main two are:

a) Couples in Accra who can afford ART might have different religions, but they are more urbane and probably more wealthy than the average couple in Ghana, and they probably have a higher level of education and access to fertility-related information (I make the assumption here that IVF is expensive in Ghana). So, the sample is varied when it comes to religion, but quite homogenous with regard to other factors. The assumption underlying the manuscript is that religion matters when it comes to these decisions and that couples interpret their religion as a way to improve their fertility, but what if that is not the case for less wealthy or more rural populations? In that case, the manuscript does only reflect a very particular slice of Ghana’s society, and that needs to be discussed and the discussion needs to be more nuanced.

As rightly pointed out the reviewer, and as indicated in lines 55-56 of the manuscript, cost of ART is very high in Ghana and it is borne wholly out of pocket. This means that it is only the upper-middle and upper classes who are able to afford. The less wealthy and the more rural dwellers irrespective of their religious positions cannot afford it. Probably if the paper were to focus on religious beliefs of both users and non-users on ART, then it would have been imperative to interview non-users as well.

However, the current paper focuses on users and how their decisions to use ART took into account their religious beliefs. Indeed, the findings show how religious beliefs influenced the choice of ART (e.g. use of donor gametes), and how some had to confer with their pastors and priests to ensure they are not breaching any religious doctrine. Moreover, considering the stigma and the psychological stress that infertile persons, especially women go through in the society, the only place where it is relatively easy to identify and recruit infertile persons is fertility hospitals. Meanwhile, the inclusion of key informants ensured that information on why some people might have refused treatment after visiting the hospital were gleaned.

In sum, our focus in this paper is not to obtain a representative sample (see lines 225-227), but to understand the experiences of infertile couples, who we admit are obviously not a homogenous category even in terms of the kinds of infertility experienced and/or in the choice of solutions utilised.

However, we understand the point of view of the reviewer as well, and we have therefore stated this possible bias in the limitations of the study. See lines 500 – 506.

b) By only choosing couples who decided to undergo ART, the authors face a selection bias: only couples whose religious feelings and communities allowed them to do ART are even interviewed, and they will obviously have a positive attitude to the subject. So, to make the conclusion that religion is a supportive factor in ART might be premature: we do not currently know what keeps other couples from attempting IVF. It might be that 95% of people reject ART based on their religion, but because the authors only interview 5%, they will not notice this in their interviews. Again, there is not much the authors can do about this, given the difficulties of getting data, but this possibility needs to be discussed and where possible, additional evidence needs to be presented to rule this out.

L.185 - I would include a brief description of what IVF and ICSI stand for, because they have different religious implication later on.

This has been provided. See lines 208 - 211

L.186 - How were key informants selected?

The selection procedure for the key informants in this study has been described. See lines 214 - 221

L.192 - The authors need to discuss their own role in this process more and potential biases that could have been introduced into the results by their own positionality. For example, later we learn that almost only women sat down for interviews, and men only as part of couples. This is worrisome and probably to do with cultural aspects and the role of the interviewer, but again this needs a nuanced discussion.

This has been done. See lines 236-245

L.215 - Does the thumbprint indicate that some participants were not literate? It feels like that is a matter worth discussing, as this could bias their responses (e.g., more dependent on community leaders to determine their attitude, more dependent on clinic staff to detail the moral implications for them).

One participant was semi-literate and preferred to thumb print rather than sign. This interview was conducted in the local language thus ruling out any issues of lack of understanding of the questions she was being asked. In addition, this particular participant is a resident of the United Kingdom having lived and worked there for more than 15 years and her decision to do the procedure in Ghana was borne out of the relatively cheaper cost associated with it in Ghana as compared to the UK. In addition, she had tried the procedure in the past year in Ghana but lost the pregnancy when she went back to the United Kingdom. She was back in the country to try it again and this time, was prepared to stay in Ghana a bit longer if she got pregnant from the procedure. 

L.227 - Could you explain what primary and secondary infertility are in this context?

This has been done. See lines 287-288

Table 1: There are many striking characteristics in this table that need to be explained in a more differentiated way, I think. First, the large age and relationship history ranges: presumably, these couples differ dramatically in why they attempt ART, how often they have tried in the past, societal pressures, and so on. The article is not specifically designed to address these questions, but presumably a couple that has been trying for pregnancy for 30 years might have a very different outlook on what is permissible than one that has been trying for 2 years, and might get very different advice from their community.

In another article, these nuances are clearly argued out to demonstrate how years of marriage and other factors such as cost of treatment, waiting and hoping etc. all work together to define the treatment seeking behaviour of these infertile couples (see Hiadzi & Boafo, 2020). As such, while the points raised are appreciated, the analysis of the data showed that, the journey towards achieving desired conception was affected by different things at different stages of the process. However, the single underlying factor that always informed the respondent irrespective of other mediating factors was the need to fulfil their societal mandate of childbearing. Interviews conducted showed that, decisions to access ART were mostly informed by cost of treatment and knowledge about the service. This present article seeks to argue how infertile couples rationalise their use of different forms of ARTs bearing in mind their religious beliefs. As such, the essence of this article is to enrich the literature on the use of ARTs from a religious lens.

Analyses and Discussions: Based on the mentioned uncertainties introduced by the method, I think the manuscript would benefit from a more detailed discussion of how representative the sentiments of the couples and their religious attitudes are for Ghana, and whether the sample selection process might have biased the interviews towards those who interpret their religion in pro-fertility way. Also, especially for the Pentecostal religions, where most couples will be converts, it is possible that participants actually joined the particular church that was most positive towards their fight with infertility and promised them to solve it.

This study is part of a bigger study on the health seeking behaviour of infertile Ghanaians in urban Ghana. While respondents who were seeking infertility treatment other than ART were asked about their religious views on ART, it was discovered that, their knowledge of ART was very limited thus their responses could not incorporated into this manuscript. 

Reviewer #2: An important question is whether the study meets the PLoS ONE data policy; the authors note limitations on the availability of the data. If those restrictions do not meet PLoS ONE data requirements, then publication criteria would preclude publishing the manuscript. Apart from that, the manuscript offers a well-structured, substantive, qualitative evaluation of a sample of Ghanians seeking and being treated with ART and how that relates to their religious beliefs. The introduction and discussion sections situate the study well, and reference both relevant findings and concepts in Ghana as well as other comparative settings. The take-away empirical points are made clearly, with illustrative quotations also useful.

The manuscript could benefit from a modest copyediting process to clean up some occasional stray commas and minor language elements, but overall the manuscript is quite readable and easy to follow. As far as other comments related to modest potential edits, please comment how common ART is in Ghana, what form(s) of ART were employed by study participants (a table in the results just points to primary and secondary fertility without providing more information), and how those services are paid for (the manuscript says in a private facility, so I am guessing this is out-of-pocket which in turn warrants brief comment on the socioeconomic backgrounds of those who can access such services).

This has been done. See lines 291-295

In addition, the forms of ART treatment employed by participants have been stated in various parts of the text namely IVF and/or ICSI

Are there any indications how representative this particular sample's findings are relative to other Ghanian ART service providers? Since all but 2 patients who were participated were women, please comment on the general ART services and demographics of those who use those services at the facility (e.g., are most patients women, or might the present sample say something more about difficulties of recruiting males who use ART?).

The reason why there was more females than males has been explained adequately in the methodology (see lines 250-261). We have also explained that the focus of this study was not to achieve representativeness. 262 – 273. 

The sample of patients only includes, by nature of the study design, those who have used ART. The study does not include patients who, perhaps by virtue of religious beliefs, elected not to receive ART. Based on the interviews with the four key informants, did they offer any insights about how commonly patients decline ART due to religious beliefs? If so, please share more from that vantage in the results.

This is similar to a point raised by reviewer 1 and has been addressed. 

However, in addition, the manuscript contains a key informant’s view that was explored about patients’ response to ART based on religious views. The manuscript reports that, according to the key informant Moslem clients out rightly refused treatment that involved masturbation to produce semen since it was against their religious beliefs. See lines 362-369

---

## [Decision Letter · Decision Letter 1]

22 Sep 2021

PONE-D-20-36012R1‘God helps those who help themselves’… religion and ART usage amongst urban GhanaiansPLOS ONE

Dear Dr. Boafo,

Thank you for submitting your manuscript to PLOS ONE. After careful consideration, we feel that it has merit but does not fully meet PLOS ONE’s publication criteria as it currently stands. Therefore, we invite you to submit a revised version of the manuscript that addresses the points raised during the review process.

Many thanks for the accurate revision of this manuscript. One of the reviewers still has some minor comments (mainly linguistic) that need to be addressed before I can make a decision about publication. 

We look forward to receiving your revised manuscript.

Kind regards,

Sara Rubinelli

Academic Editor

PLOS ONE

Journal Requirements:

Reviewers' comments:

Reviewer's Responses to Questions

**Comments to the Author**

1. If the authors have adequately addressed your comments raised in a previous round of review and you feel that this manuscript is now acceptable for publication, you may indicate that here to bypass the “Comments to the Author” section, enter your conflict of interest statement in the “Confidential to Editor” section, and submit your "Accept" recommendation.

Reviewer #2: (No Response)

2. Is the manuscript technically sound, and do the data support the conclusions?

Reviewer #2: Yes

3. Has the statistical analysis been performed appropriately and rigorously? 

Reviewer #2: N/A

4. Have the authors made all data underlying the findings in their manuscript fully available?

Reviewer #2: Yes

5. Is the manuscript presented in an intelligible fashion and written in standard English?

Reviewer #2: Yes

6. Review Comments to the Author

Reviewer #2: In this resubmission, the authors have provided more information about the sociodemographic characteristics of the sample, which is important with respect to the question of generalizability. The authors have addressed some of the other points raised by the other reviewer and myself. I do not advocate for the necessity of further substantive edits. However, the manuscript has some typos and awkward phrasings that warrant copyediting. Some examples include: a) capitalizating "catholic" as "Catholic"; b) lines 249-251 and thereabouts: there is some redundancy and errant phrasing concerning "they often appeared economical"; c) line 267: "lot of time on hands to while away" could be shortened to something like "had free time available"; d) around line 518 and subsequently: the limitations passage is a bit awkward such as "this study does not seek to make generalize..." plus the key point could still be clearer that it is unclear how findings from individuals not seeking ART might compare with those in the manuscript among ART users at even the very facility (e.g., no data on whether religious background impacts perceived "uptake" of services even at the single recruitment facility). This all said, if such copyedits can be made then as noted previously I believe this manuscript makes a qualitative contribution to its purported topic, and does so with interesting and well-presented empirical details (e.g., Results).

7. PLOS authors have the option to publish the peer review history of their article (what does this mean?). If published, this will include your full peer review and any attached files.

Reviewer #2: No

---

## [Author Response · Author response to Decision Letter 1]

7 Oct 2021

Reviewer #2: 

a) capitalizating "catholic" as "Catholic"; 

 This has been done. See lines 215 and 216. 

b) lines 249-251 and thereabouts: there is some redundancy and errant phrasing concerning "they often appeared economical"; 

 The entire sentence in line 244 and 245 has been deleted thus correcting the redundancy and errant phrasing

c) line 267: "lot of time on hands to while away" could be shortened to something like "had free time available"; This has been done

d) around line 518 and subsequently: the limitations passage is a bit awkward such as "this study does not seek to make generalize..." 

 This has been corrected to read…..’this study does not seek to generalize…’

e) the key point could still be clearer that it is unclear how findings from individuals not seeking ART might compare with those in the manuscript among ART users at even the very facility (e.g., no data on whether religious background impacts perceived "uptake" of services even at the single recruitment facility). 

 This has been done to make the point clearer. See lines 504-506

Copy editing on the manuscript has been done with all typos and awkward phrasings corrected.

---

## [Decision Letter · Decision Letter 2]

9 Nov 2021

‘God helps those who help themselves’… religion and Assisted Reproductive Technology usage amongst urban Ghanaians

PONE-D-20-36012R2

Dear Dr. Boafo,

We’re pleased to inform you that your manuscript has been judged scientifically suitable for publication and will be formally accepted for publication once it meets all outstanding technical requirements.

Kind regards,

Sara Rubinelli

Academic Editor

PLOS ONE

Additional Editor Comments (optional):

Reviewers' comments:

Reviewer's Responses to Questions

**Comments to the Author**

1. If the authors have adequately addressed your comments raised in a previous round of review and you feel that this manuscript is now acceptable for publication, you may indicate that here to bypass the “Comments to the Author” section, enter your conflict of interest statement in the “Confidential to Editor” section, and submit your "Accept" recommendation.

Reviewer #2: All comments have been addressed

2. Is the manuscript technically sound, and do the data support the conclusions?

Reviewer #2: (No Response)

3. Has the statistical analysis been performed appropriately and rigorously? 

Reviewer #2: (No Response)

4. Have the authors made all data underlying the findings in their manuscript fully available?

Reviewer #2: (No Response)

5. Is the manuscript presented in an intelligible fashion and written in standard English?

Reviewer #2: (No Response)

6. Review Comments to the Author

Reviewer #2: (No Response)

7. PLOS authors have the option to publish the peer review history of their article (what does this mean?). If published, this will include your full peer review and any attached files.

Reviewer #2: No

---

## [Editor Report · Acceptance letter]

1 Dec 2021

PONE-D-20-36012R2 

‘God helps those who help themselves’… religion and Assisted Reproductive Technology usage amongst urban Ghanaians 

Dear Dr. Boafo:

I'm pleased to inform you that your manuscript has been deemed suitable for publication in PLOS ONE. Congratulations! Your manuscript is now with our production department. 

Kind regards, 

on behalf of

Dr. Sara Rubinelli 

Academic Editor

PLOS ONE